# Representation of molecular structures with persistent homology for machine learning applications in chemistry

Jacob Townsend[1], Cassie Putman Micucci [2], John H. Hymel [1], Vasileios Maroulas [2✉] & Konstantinos D. Vogiatzis [1✉]

Machine learning and high-throughput computational screening have been valuable tools in accelerated first-principles screening for the discovery of the next generation of functionalized molecules and materials. The application of machine learning for chemical applications requires the conversion of molecular structures to a machine-readable format known as a molecular representation. The choice of such representations impacts the performance and outcomes of chemical machine learning methods. Herein, we present a new concise molecular representation derived from persistent homology, an applied branch of mathematics. We have demonstrated its applicability in a high-throughput computational screening of a large molecular database (GDB-9) with more than 133,000 organic molecules. Our target is to identify novel molecules that selectively interact with $CO_2$. The methodology and performance of the novel molecular fingerprinting method is presented and the new chemically-driven persistence image representation is used to screen the GDB-9 database to suggest molecules and/or functional groups with enhanced properties.

[1] Department of Chemistry, University of Tennessee, Knoxville, TN 37996-1600, USA. [2] Department of Mathematics, University of Tennessee, Knoxville, TN 37996-1320, USA. ✉email: vmaroula@utk.edu; kvogiatz@utk.edu

The increasing concentration of greenhouse gases has been identified as a primary factor of many facets of environmental degradation such as higher global temperature, rising sea levels, increased ocean acidity, and more extreme weather-related events. $CO_2$ is the most prominent greenhouse gas, and its atmospheric concentration has exceeded 400 ppm, which is more than a 40% increase from pre-industrial conditions, potentially leading to a rise in global temperatures of more than 2 °C by the year 2100[1]. Lowering $CO_2$ emissions is therefore mandatory to meet ambitions to limit temperature increases to 1.5 °C[2]. Advancements in carbon capture and storage technology are desired for meeting these goals of lowered atmospheric greenhouse gas emissions and reduced global temperature increases year to year. At an industrial level, liquid amine-based solvents are used for separation and capture of $CO_2$ via chemisorption, but the solvent regeneration step is an energy intensive process. Membrane-based technologies offer an alternative, cost-effective process for $CO_2$. Unlike solvents, where chemisorption involves a reaction with binding strengths exceeding 20 kcal mol$^{-1}$ through the creation of chemical bonds between $CO_2$ and solvent, membranes utilize much weaker noncovalent interactions. Different types of materials have been suggested for the fabrication of permeable membranes including amorphous, non-porous polymeric membranes[3–6], or crystalline materials with permanent porosity such as metal-organic frameworks (MOFs) or zeolites[7].

The understanding of how the atomistic structure of materials affects the gas selectives is a crucial process for the development of more efficient carbon capture technologies. Most often, this involves the separation of $CO_2$ from $N_2$, which are two atmospheric gases with similar kinetic diameters, making size-sieving a challenging task. The introduction of functional groups which selectively interact with $CO_2$ has been a successful approach for increasing membrane performance[6,8]. Such $CO_2$-philic functional groups (usually Lewis bases) can be either introduced into the framework of porous crystalline materials (e.g., MOFs) or functionalized into the repeat units of non-porous polymeric membranes. Electronic structure theory calculations between molecular units and the respective gases provide a quantification of these noncovalent interactions, as well as elucidate their nature and properties[9–15]. However, the number of potential $CO_2$-philic groups is intractably large, which leads to an excessive study of such systems with accurate ab initio methods. In addition, the determination of gas interaction energies may require multiple calculations to evaluate competitive gas binding sites for every structure, which further increases the computational cost and expert intervention. However, high-throughput computational screening can accelerate the discovery of new, functional materials for rational synthesis through the circumvention of the expensive and time-demanding synthesis and testing process[16,17].

High-throughput computational design has shown great success in identifying new molecules[18,19] and materials[20–23] with enhanced properties and advanced functionality. For many applications, first-principles studies are essential to virtual screening, but the high computational cost of these methods makes the search of large parts of the chemical space cost-prohibitive. In recent years, machine learning (ML) has become a valuable tool in reducing the cost of a systematic chemical space exploration by enhancing the search for structure-property relationships[24–26], guiding molecular design[27–30], and predicting electronic structure properties[31–35]. ML algorithms are used for their ability to learn complicated relationships in data with high computational efficiency that can be systematically improved through additional training data, but may require extensive training set sizes before predicting out-of-sample properties accurately. The efficiency of ML depends on how these data are passed to the algorithm. For chemical applications, this occurs through molecular representations, which are the featurization of molecular compounds from their molecular structure into a vector of values. The ML algorithm then infers the relation between the structure and the property of interest. Recent developments in the formulation of molecular representations, particularly in the realm of quantum properties and structure-function relations, have increased the efficiency of ML for chemical applications[36–42]. In addition, such representations generalize to more complex instances such as reaction barriers[43]. Despite the inherent ability of ML to extract important features, ML-model accuracy is dependent on the molecular representation.

A molecular representation reduces the dimensionality of a molecular structure into a chemically meaningful format that relays important chemical information. For example, a chemical formula conveys a three-dimensional molecule as a string of characters but it is an ambiguous input for ML. Atomistic and molecular structure should be converted into a machine-readable format that can be parsed efficiently to ML algorithms as input features. One of the most prominent representations is the Coulomb matrix (CM) introduced by Rupp et al., which is a square atom-by-atom matrix containing an approximate potential energy of the free atom along the diagonal and pair Coulombic potentials on the off-diagonal terms[33]. An improvement over CM is typically observed using the Bag-of-Bonds (BoB) representation, where each atomic pair is placed in specific vectors (bags) based off the elemental pairs and sorted by value[34]. Faber et al. have developed FCHL, a representation based on Gaussian distribution functions for the universal kernel ridge regression-based quantum machine-learning models[42]. In addition, the smooth overlap of atomic positions (SOAP) representation[44] calculates the local density of atoms around all atoms in a given chemical environment but suffers from an increased computational cost over the pairwise CM and BoB representations.

Herein, we present a new molecular representation scheme based on persistent homology, a branch of computational topology. Application of persistent homology on molecules encodes three-dimensional structural data into two-dimensional persistence images. Since persistence images hold topological features of chemical structures, we are suggesting them as alternative molecular fingerprints which are transposed into ML input and used to identify relationships in the data. A molecular representation is introduced for encoding chemical structures, which is applied in the prediction of interaction energies of organic molecules with gas molecules. Persistence images offer a similar-size ML vectorization regardless of system size, and a numerical example is given as a proof of this concept. Whereas CM and BoB do not provide a constant-size representation by this definition, it is effectively achieved through padding empty cells. However, the input vector space takes the dimensions of the largest molecule in the data and requires significant padding for smaller molecules, whereas the introduced method has a predefined vector size regardless of number of atoms. It has been suggested that using feature vectors with sizes independent of system size may result in improved generalization between small and large systems[45]. The new method is used for screening a large database of organic molecules for the discovery of $CO_2$-philic functional groups.

## Results

**Persistence diagrams**. The mapping of molecular structure to a chemically driven persistence image entails several steps. First, the molecular homological features, which measure the connectedness, proximity, and the empty space among the atoms, are computed and stored. These homological features are summarized in a persistence diagram (PD)[46–58]. A PD encodes molecular features such as bonds and rings. The PDs can then be vectorized

into a persistence image[59] (PI) for use as a molecular representation. However, employing persistent homology and its derivative, PI, purely focuses on detecting topological attributes but lacks explicit incorporation of key chemical information such as element identity, leading to limited applicability in molecular systems. Here, we describe the application of persistent homology with domain-specific knowledge for the generation of persistence images based on atomic properties. The basic steps are presented in the following paragraphs. Anisole was selected as a representative example because it contains two distinct functional units (phenyl- and methoxy groups).

To construct a persistence diagram for a given molecule, spheres of a given radius centered at each atom are considered and, as the radius increases, the spheres intersect and lead to the evolution of homological features, called connected components and holes. The connected components encode interatomic distances, while holes describe molecular attributes such as rings and functional groups.

The PDs hold information about the generation or birth and the lifetime length or persistence of connected components and holes. The placement of a birth is denoted by its location along the $x$-axis of the PD, whereas the persistence is denoted by its location on the $y$-axis. Birth of connected components occurs at 0, since every atom is given an initial sphere with radius 0 at the start of the algorithm (see anisole as an example on Fig. 1). The spheres are then systematically expanded (Fig. 1a, d, g, j, and m) until spherical intersections occur, which effectively generate a new connected component by merging older ones. The

persistence of connected components is then recorded on the associated PD. In some sense, a PD records the time in terms of spheres' radii for the atoms to form a single cluster. For anisole that is used here as an example, four different types of connected components are generated. The first two appear at ~1.1 and correspond to the C–H bonds of the methoxyl and phenyl groups, respectively. The other two appear at ~1.4 and correspond to the C–O and C–C bonds, respectively. This means that the units of the two axes are given in angstroms (Å). When sphere intersections lead to the formation of connected atoms (connected component) on a ring, for example when all the six spheres of the phenyl carbons have met, a hole is generated (Fig. 1l). The death of a hole occurs when all spheres that form a given hole intersect, and its persistence is recorded on the persistence diagram (Fig. 1o).

It becomes now evident that the connected components depend on the distances of neighboring atoms and the holes correspond to topological features of the functional groups. Two holes are now formed for the anisole example of Fig. 1, which correspond to the phenyl and methoxy groups. These holes are unique for each respective unit and differentiate between different conformations of the same molecule and subtle differences in geometry[60–63].

**Chemically driven persistence images as molecular representations**. The PD is vectorized into a pixelized image, called persistence image (PI), which is a stable, computationally

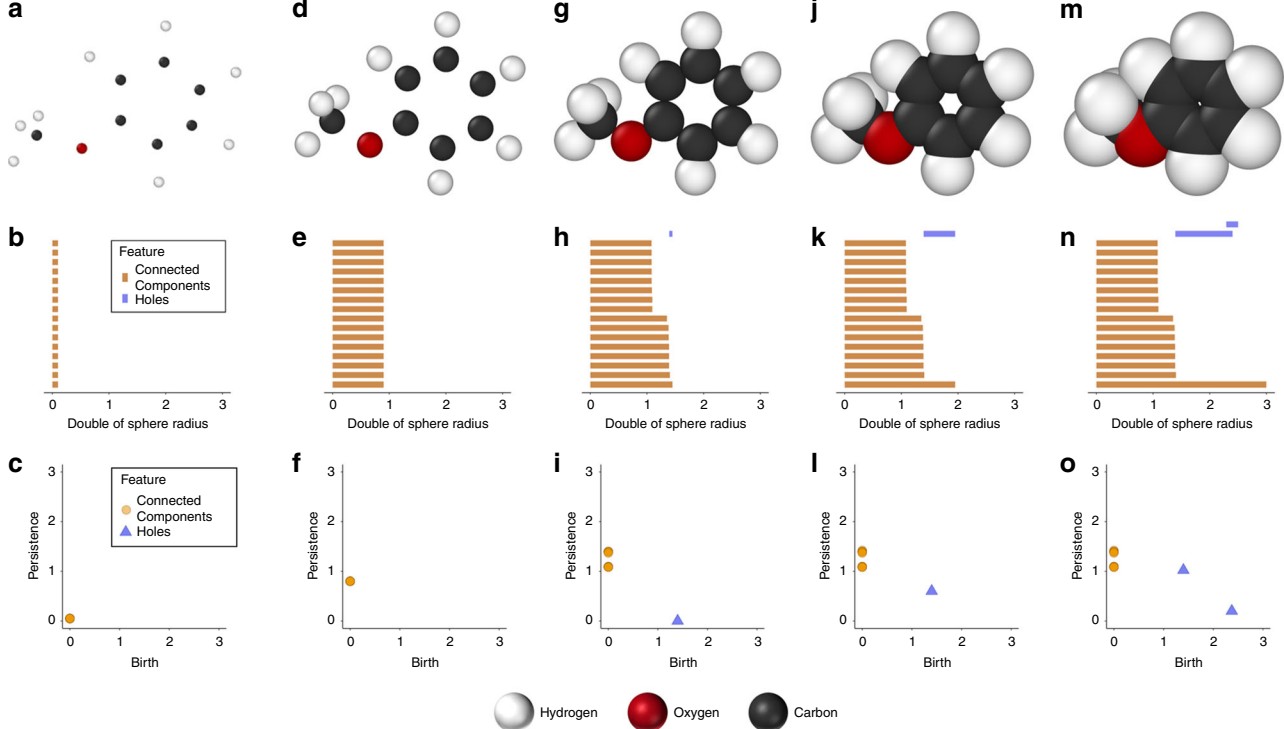

**Fig. 1 Graphical representation of the evolution of the persistent diagram (PD) for anisole. a** Each atom of anisole is represented by a point. Each connected component is born at time birth = 0 in the evolution of **b** homological features plot and **c** PD. Each bar at the evolution plot tracks the connected components or holes. Note that there are 16 overlapping points in the PD associated with the number of atoms. **d–f** The radius of a sphere centered at each atom is increased, and the connected components continue to persist so the persistence of a connected component is not in its final location. **g–i** When the radius increase leads to sphere overlap between atom pairs, the connected components on the PD are finalized. A hole is created by the six carbon atoms of the phenyl group, and a blue triangle appears in the persistence diagram to indicate its birth. **j–l** The radius is again expanded, and the hole (phenyl ring) still persists. **m–o** A second hole is formed between the methoxyl group and the two carbons of the phenyl group. The death of both holes occurs when their corresponding spheres intersect. The final PD is obtained when the death of all but one connected component and all holes are observed. Note that the bottom bar never terminates and it is always excluded from the PD.

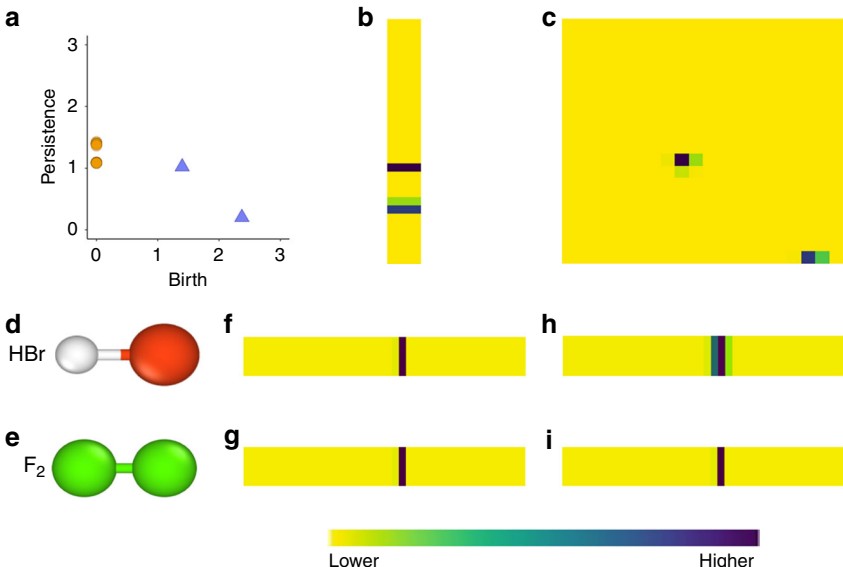

**Fig. 2 Generation of a persistence image (PI) from a persistence diagram (PD). a** PD of anisole, as shown in Fig. 1. The PI for **b** connected components and **c** holes. The molecules **d** HBr and **e** $F_2$ have the same bond length, resulting in equivalent features on the persistence diagram. Without incorporating the chemically driven PI, the representations in **f** and **g** are equivalent. By incorporating the pairwise electronegativity difference into the variance of the Gaussian kernel, the input vectors **h** and **i** for the two molecules are distinguished by the variance of the vector for HBr.

tractable representation[59]. PIs are constructed by placing a Gaussian kernel centered at each point on the PD as it is high-lighted for anisole in Fig. 2, where the pixel intensity corresponds to the multiplicity. For example, five C–H pairs in the phenyl group, three C–H pairs in the methyl group etc. Next, the surface is transferred into pixel values (Fig. 2b, c). The resulting image effectively encodes the molecular geometry.

The transformation of a PD to a PI may lead to inconsistencies if all atoms are treated identically, especially when molecular structures with the same geometries but different atom types are encoded into a PI. For example, the diatomic molecules HBr and $F_2$ have approximately the same bond distance (1.41 Å), and therefore generate the same PI (Fig. 2f, g, respectively). To ameliorate this shortcoming, we introduce atomistic information in the variance of the Gaussian kernels that yielded a PI. The variance determines the spread of the kernel, or how "smeared" each point on the PD is when placed onto the PI. This variance is chosen based on the atom type that created the point in the PD. Specifically, we define the variance in the persistence images by the difference in electronegativity for connected components. Electronegativity differences are chosen because they provide a general description between the nature of different bonds. For the example of HBr and $F_2$, HBr has a very polar chemical bond, whereas molecular fluorine is nonpolar. Large variance is provided to atom pairs with large electronegativity differences, which ultimately generates unique PIs (Fig. 2h, i, respectively). Our new chemically driven persistence image differentiates between molecules which have similar geometric configurations but different atomic compositions.

Another important feature is related to the dimensionality of the PI molecular representation which remains of the same order with respect to the molecular size as we show empirically in Supplementary Figs. 2 and 3. This is what we call herein a similar-size representation. For example, for a small molecule like anisole that is composed of 16 atoms, a 3 Å × 3 Å PD was generated. The equivalent PD of a medium-size molecule such as the *tert*-butylcalixarene (105 atoms) is of comparable size (4 Å × 4 Å). Similarly, the PD of a large structure, the main protease of the new coronavirus identified as COVID-19[64] in complex with an

inhibitor N3 (2500 non-hydrogen atoms) has size of 6 Å × 6 Å. A detailed analysis is given in Supplementary Note 2. As it was mentioned in the introduction, such a similar-size representation is desirable for many chemical applications[45].

**Performance of persistence images as molecular representations**. Here, we demonstrate the performance of the chemically driven PIs on an application relevant to green chemistry. Our aim is to screen a large molecular database in order to discover molecular groups that show a stronger affinity for $CO_2$ interaction over $N_2$. Such molecular groups can be introduced in polymeric materials for the development of the next generation of functional gas separation membranes. Since it is desirable to avoid any density functional theory (DFT)-optimized geometries as input for ML models, which introduce a significant computational bottleneck for the screening of large molecular databases, we resort to structures generated by the OpenBabel[65] software package (gen3d function). Our target is to train a ML model that maps low-cost geometries with accurate quantum chemical data, so it can provide reliable interaction energies for molecular species with geometries generated on-the-fly.

We tested the performance of PI as alternative molecular representations that effectively encode chemical structures. The initial subset of 100 organic molecules was used, generated based on the procedure described in Supplementary Note 3. The interaction energies of each of these structures with $CO_2$ and $N_2$ were computed by means of DFT. We also wanted to compare PI with the widely utilized Coulomb matrices (CM), Bag-of-Bonds (BoB), FCHL, and Smooth Overlap of Atomic Positions (SOAP) representations, since each of these are produced with little computational burden and widely implemented in a number of programming libraries[66–68]. PI, CMs, BoBs, FCHL, and SOAP representations were generated for each structure and the performance of each representation scheme was evaluated for the prediction of gas interaction energies. For each scheme, a variety of machine-learning algorithms were tested, including random forest, Gaussian process regression, and kernel ridge regression. A detailed analysis of the optimization process is given in Supplementary Note 6. Overall, two machine-learning models

were trained per molecular representation scheme, one for $CO_2$ and one for $N_2$ interaction energies. The 10-fold cross validation root-mean-squared error (RMSE) for the five trained models on the $CO_2$ energies are shown in Fig. 3. The error bars represent the standard deviation of the RMSE. Similar results were obtained for $N_2$ interaction energies (see Supplementary Note 7). Comparing the best learners for each representation, CM showed the highest deviation (RMSE of 0.63 kcal mol$^{-1}$), followed by BoB and FCHL (RMSE of 0.52 and 0.50 kcal mol$^{-1}$, respectively). The most accurate models were PI with kernel ridge regression (Laplacian kernel, 0.44 kcal mol$^{-1}$) and SOAP with kernel ridge regression (linear kernel, 0.41 kcal mol$^{-1}$), where PI showed a tighter variance, yielding a higher confidence in the predictions.

**Screening the GDB-9 database**. High-throughput computational screening using ML is an efficient method to survey molecules for numerous chemical applications. Here, we are applying the PI method for identifying molecules and functional groups that enhance $CO_2$ interactions with little computational cost. ML models trained on DFT-quality data can estimate DFT-quality results for hundreds of thousands of systems within seconds, while the explicit computation at the DFT level is a cost-prohibitive process. The GDB-9 database[69] was screened, which includes 133,885 organic molecules containing no more than nine

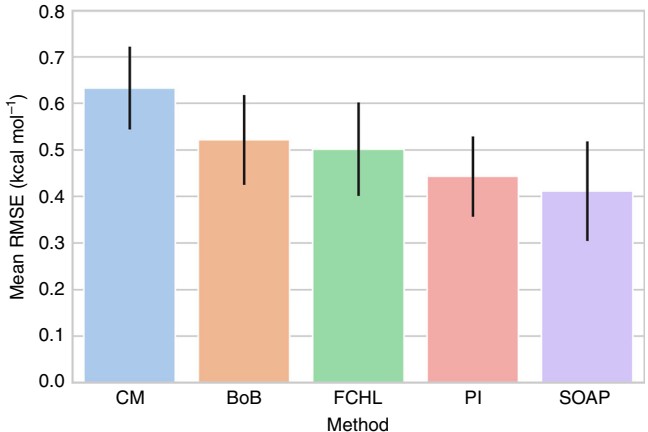

**Fig. 3 Root-mean-square error (RMSE, in kcal mol$^{-1}$) of $CO_2$ interaction energies.** The automated OpenBabel-generated structures were used for each molecular representation model. The error bars represent the standard deviation of the RMSE.

non-hydrogen atoms to determine the most promising molecules for $CO_2$ binding.

The data from the initial 100 organic molecules discussed in the previous section do not adequately capture the properties of the chemical space spanned by the GDB-9 database. The initial training set can be considered as biased since it contains largely N-containing heterocycles with small functionalizations on aromatic carbons (see Supplementary Note 9). For surpassing this limitation and reliably screening the full GDB-9 space, we applied a methodology known as active learning. In active learning, the training set is systematically expanded to capture the necessary missing physics to accurately predict for the targeted space. The top 40 molecules were selected with respect to predicted $CO_2$ interaction strength and further investigated by the MD/DFT scheme described in the computational details (Supplementary Note 4). Therefore, the training set was expanded to better infer the relationship between the molecular representation and the chemical space spanning the GDB-9 database. We have repeated this processes four times by considering different yet optimized molecular representation methods (CM, BoB, SOAP, and PI). The individual steps that were followed are shown schematically in Fig. 4 and analyzed in the next paragraphs.

Three iterations were performed with each representation scheme together with the optimized machine-learning algorithm, as it is discussed in the previous section and in Supplementary Note 6. Thus, the kernel ridge regression (Laplacian) was used for CM and PI, Gaussian process regression for BoB, and kernel ridge regression (linear) for SOAP. The active-learning process resulted a total of 220 data points, i.e., 220 molecular structures with their corresponding $CO_2$ and $N_2$ interaction energies computed by DFT per method. The distributions of the interaction energies of these molecules for each method are visualized in Fig. 5. We set a mark at $-6.0$ kcal mol$^{-1}$ for molecules with significantly strong $CO_2$ interaction energy. For a detailed analysis of the mean and median of each active-learning iteration per method, we refer the reader to the Supplementary Note 8. The first iteration contains only the original 100-molecule training set. By expanding the training set with the 40 molecules from the first iteration, the next 40 best-predicted molecules from the model that utilizes the PI representation have significantly improved. On the contrary, no significant changes were observed from the other three models. By the third iteration, the $CO_2$ distribution from CM, BoB, and PI remained almost unchanged, while a small shift toward stronger interaction energies was found for the SOAP model. Overall, PI showed the greatest performance since each respective iteration increased the number of promising structures, from 10 (first

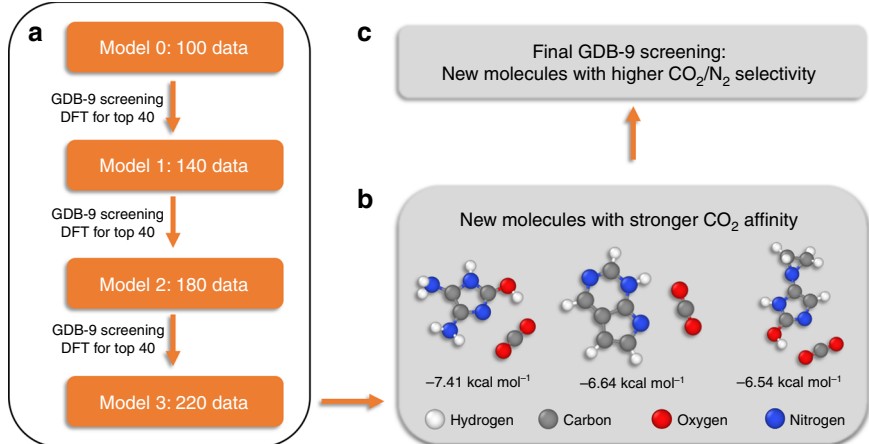

**Fig. 4 Flowchart of the intermediate steps followed for the high-throughput computational screening of the GDB-9 database. a** The three active-learning steps starting from a model with 100 molecules (Model 0), **b** the results obtained from DFT, and **c** the screening of the GDB-9 database.

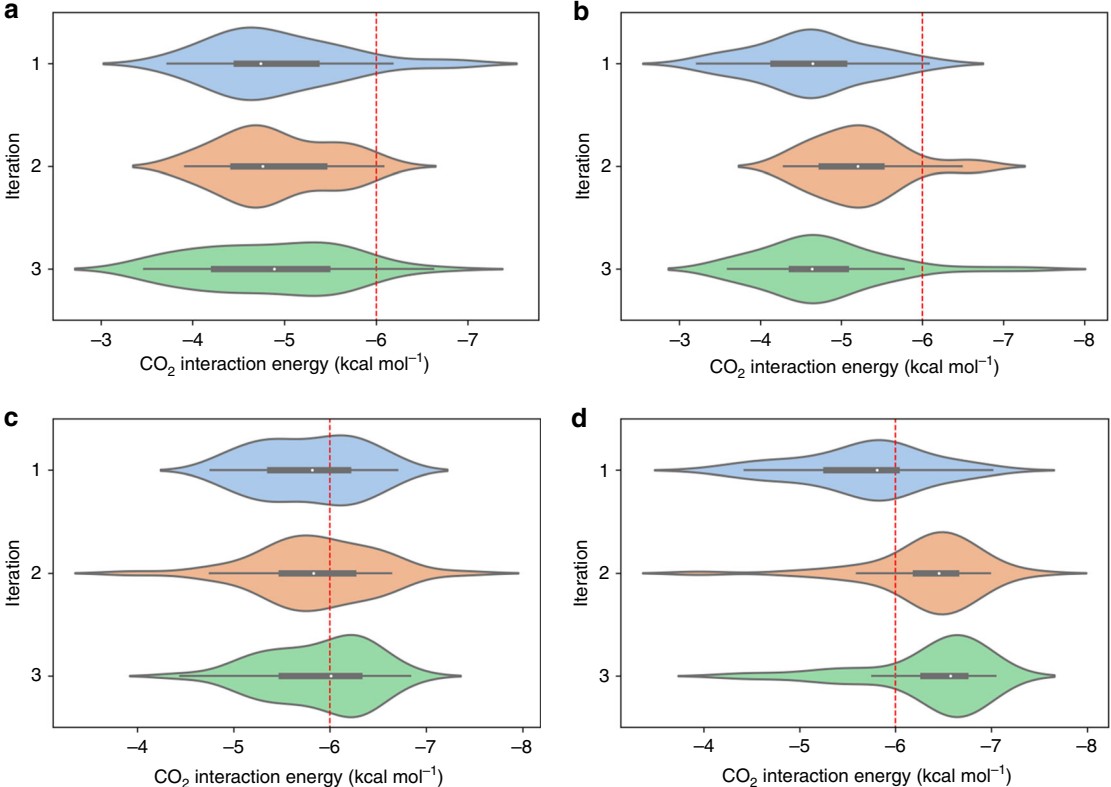

**Fig. 5 CO$_2$ interaction energy distribution shown as horizontal violin plots for the first, second, and third active-learning steps.** The height of the shape shows the frequency of occurrences, the white dot along the axis iteration depicts the median, and the bold line the middle 50% of the data. The thinner line corresponds to the range of all the data. The dashed line at −6.00 kcal mol$^{-1}$ marks cases with stronger CO$_2$ interaction.

iteration) to 43 (second iteration), and ultimately to 75 out of 120 molecules. Active learning with the PI molecular representation has systematically expanded the training set to better represent the chemical space of the dataset, yielding more reliable predictions every round. Since promising structures are rare within the dataset, this strategy allows the model to account for these rare instances within the training set in a way that would be impossible with a randomly chosen training set. In addition, the three top candidates for CO$_2$ separations were found to demonstrate stronger interaction energies than −6.50 kcal mol$^{-1}$, which are shown in Fig. 4 (bottom, right). Our computational procedure allowed us to discover new molecules with higher CO$_2$ affinity that combine previously unknown binding motifs. In particular, we found that cooperative effects between N-containing heterocycles with amino or hydroxo groups at ortho position increases the CO$_2$ strength. The lone electron pair of nitrogen 6induces a dipole moment on CO$_2$ that allows stronger interactions with hydrogen atoms of the NH$_2$− and/or OH− functional groups.

After completing three iterations of active learning, the full dataset (220 molecules) is used to create a ML model for predictions on the whole GDB-9 database. For comparison, the database was screened with the four different models, where each of them uses a different molecular representation method (CM, BoB, SOAP, and PI) and data generated from the corresponding active-learning steps. The optimum learner for each method was used, as it was discussed in the previous section, except for BoB, where the kernel ridge regression (linear) was applied (for a detailed discussion, see Supplementary Note 8). Figure 6 includes a plot for each model, where all predicted N$_2$ and CO$_2$ interaction energies are on the $x$- and $y$-axis, respectively. Only the method that utilized the PI molecular representation was able to identify 4,5-diamino-1H-

imidazol-2-ol as one of the molecules with the strongest CO$_2$ affinity as indicated with an orange dot on each plot of Fig. 6. 4,5-diamino-1H-imidazol-2-ol was part of the training set introduced in the second step of the active-learning process (Fig. 4), and has a DFT CO$_2$ interaction energy of −7.41 kcal mol$^{-1}$. All methods agree that the majority of the molecular entries of the GDB-9 dataset have a mean CO$_2$ interaction energy centered between −3.0 and −4.0 kcal mol$^{-1}$ and a N$_2$ interaction energy at −2.0 kcal mol$^{-1}$. Most of the molecules have predicted CO$_2$ interaction energies between −3.0 and −5.0 kcal mol$^{-1}$, which emphasize the difficulty in determining new molecules with high CO$_2$ affinity. However, CM and BoB were less effective in identifying rare instances. On the contrary, results from SOAP were significantly scattered, and predicted many cases with false CO$_2$ interaction energies close to 0 kcal mol$^{-1}$ (in a few cases, SOAP even predicted repulsive interaction energies, Supplementary Note 8). Interestingly, training of machine-learning models with the CM, BoB or the SOAP representations that utilize the molecules identified by active learning with the PI representation yielded more concise distributions, while all models identified the molecular species with the stronger CO$_2$ interaction energy among the top candidates (Supplementary Fig. 7). In other words, PI provided higher quality data for all methods. Therefore, the model that utilizes PIs for molecular representations provided the most consistent distributions for both CO$_2$ and N$_2$ interaction energies. The PI screening revealed a total of 44 of the 133,885 molecules with CO$_2$ interaction energies exceeding −6.5 kcal mol$^{-1}$. DFT calculations were performed for verification of these results. It should also be mentioned that SOAP needed 88,287 s for screening the full GDB-9 database, while the screening with the novel PI representation was almost 40 times faster (only 2219 s). All screenings were performed on an Intel® i5-4278U processor.

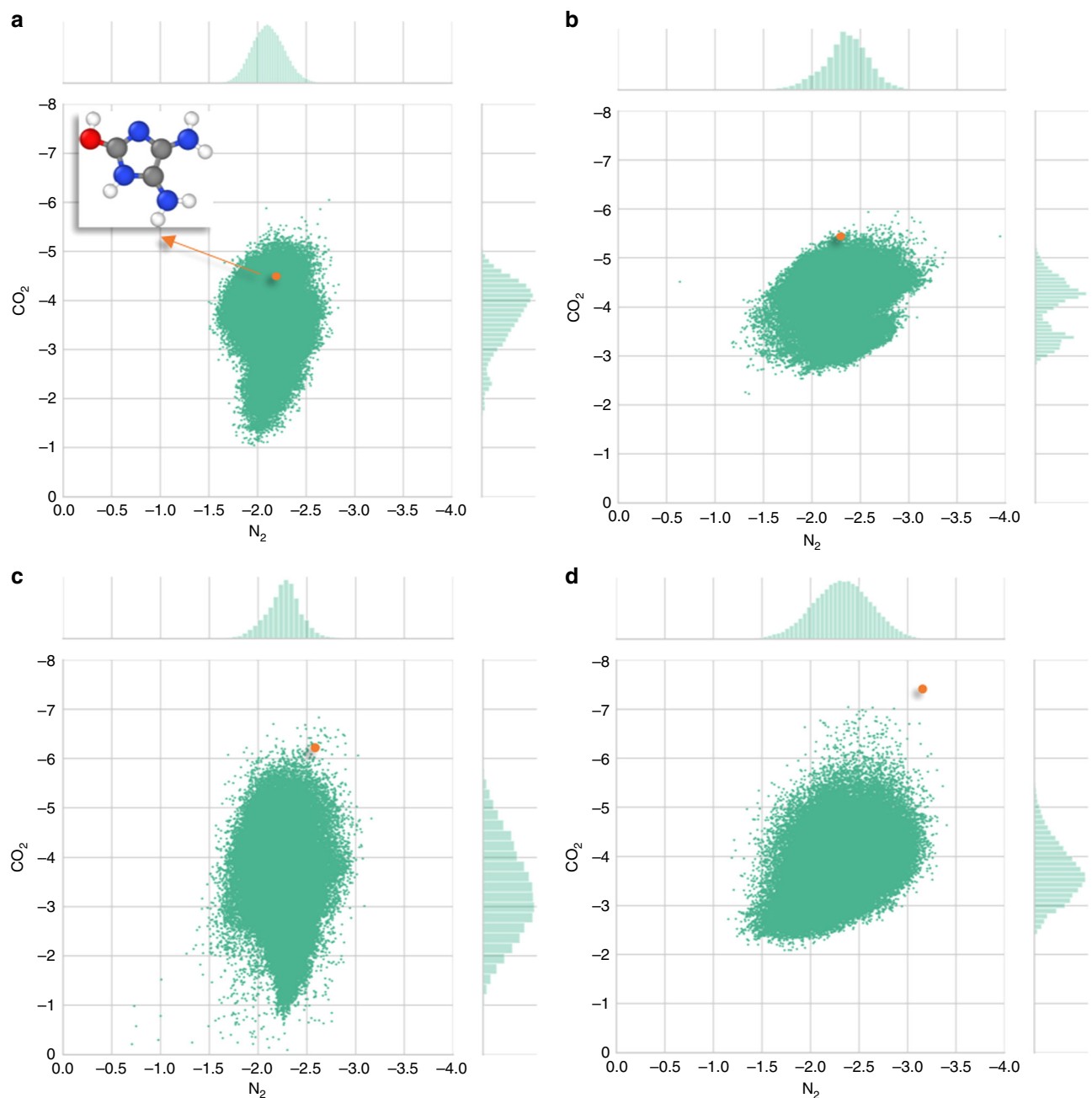

**Fig. 6 Predicted CO$_2$ and N$_2$ interaction energies (in kcal mol$^{-1}$) for all molecules in the GDB-9 database using four molecular representation models.**
Only the model that utilized the PI molecular representation was able to identify 4,5-diamino-1H-imidazol-2-ol (inset) as the strongest CO$_2$-philic groups (shown with an orange dot on each plot).

## Discussion

A novel molecular representation utilizing persistence images with embedded chemical bonding information has been introduced for predicting DFT-quality CO$_2$/N$_2$ interaction energies. From our investigation, this new chemically driven persistence image is a concise, computationally efficient, and effective representation that generally outperforms other representations for prediction of CO$_2$ interaction energies since the computational cost is low and does not suffer from dimensionality problems. This representation accounts for underlying topological structure in the molecule, providing a method to control uncertainty due to differing geometric configurations. The new methodology has been applied for the screening of the GDB-9 database to suggest new CO$_2$-philic moieties. By using an active-learning approach, our ML-based screening was able to identify many promising molecules in the GDB-9 database despite a very small training set (220 molecules). Specifically, 44 molecules were identified that exceed −6.5 kcal mol$^{-1}$ CO$_2$ interaction energy. In addition, candidates that may exhibit strong CO$_2$ interactions while maintaining weak N$_2$ interactions were examined, yielding a strategy for identifying species with potentially strong gas separation capabilities. Ultimately, chemically driven persistence images are promising molecular representations for larger supermolecular systems due to compact vectorization. Therefore, we believe that the chemically driven PI molecular representations can be applied in a plethora of chemical problems.

The PI method described herein relies on a topological representation of a molecular compound that allows a flexible summary

of the diversity of the atomic geometries. Due to this flexibility in terms of topological equivalence, the PI is robust and provides accurate predictions in contrast to other methods that need to learn rigid geometric representations. We are currently expanding the applicability of the novel molecular fingerprinting method to high-throughput screening of molecular databases for catalysis and ligand-based lanthanide/actinide separations. For this type of chemical applications, additional features are taken into consideration, such as intensity normalization when a PI is generated from the corresponding PD and predictability of properties of larger molecules from data generated from smaller ones.

## Data availability
The data for the high-throughput computational screening are available in the Supplementary Information.

## Code availability
The code for the numerical simulations is available at https://gitlab.com/voglab/PersistentImages_Chemistry.

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

## Acknowledgements

This material is based on work supported by the National Science Foundation under Grant CHE-1800237 (J.T., J.H.H., and K.D.V.), and by the ARO Grant # W911NF-17-1-0313 and the NSF DMS-1821241 (C.P.M. and V.M.). This work used the Advanced Computer Facility (ACF) of the University of Tennessee.

## Author contributions

V.M. and K.D.V. conceived the project. J.T. and C.P.M. wrote the code. J.T. and J.H.H. performed the calculations. J.T., C.P.M., V.M., and K.D.V. wrote the paper.

## Competing interests

The authors declare no competing interests.
