## [Peer Review File · Nature Communications]

Reviewers' comments:

Reviewer #2 (Remarks to the Author):

The authors present a molecular representation, which they consider to be easy to calculate and of the same size across all molecules (size-consistent). The authors apply it to train the CO₂/N₂ interaction energies of organic molecules within the GDB-9 set. The authors claim that this representation generally outperforms other representations.

The simplicity of the descriptor is very appealing and it appears to provide performance that is en par with the, by now outdated, coulomb matrix and BOB descriptors, but I do not see how the claim of 'generally outperforming other representations' is sufficiently substantiated in the manuscript.

-Within the error bars provided in Figure 3, BOB and PI show the same performance for a very small set of 100 interaction energies. I do not find this convincing. For such a small set of molecules, the authors could have systematically assessed a number of more state-of-the-art representations, such as FCHL, Behler-Parrinello, SOAP, or SchNet (not a descriptor per se, but a comparable approach). This would also have provided an opportunity to showcase the computational efficiency of PI, which is emphasized in the manuscript, but not quantified.

-The training protocol based on a series of active learning steps is well designed and seems appropriate for the envisioned application. As it only requires a small number of molecules, I wonder why the authors have not also trained a model for CM, BOB and PI to compare the methods within Figure 6. This would have provided for a more robust benchmark.

- I worry about the significance of the findings in Figure 6 and the CO₂/N₂ interaction energy ratios that are discussed. All N₂ interaction energies fall within a range of 1 kcal/mol, which is somewhat below the trust radius of PBE0-D3. The authors select PBE0-D3 based on a benchmark of CO₂ interaction energies against coupled-cluster results, but it is not clear from the manuscript or SI if this level of confidence also applies to N₂.

- It is not clear what the benefit of a 'size-consistent' descriptor is. As the authors state, all descriptors can be padded to be of consistent size for molecules of different composition. The authors do not provide evidence for cases in which there would be a material benefit to size-consistency. How is this property particularly beneficial in the proof-of-principle application that they discuss? Figure 2 also shows that the range that is covered in the pixelated diagrams that correspond to the final descriptor depends on the size of the molecule (no. of births and level of persistence can vary). This means that the size of the Birth/Persistence axis is controlled by the largest and most complex molecule in the dataset. One could consider this to be a form of zero-padding when the same range is applied to smaller/less complex molecules.

Reviewer #3 (Remarks to the Author):

The manuscript introduces persistent homology-based method for transforming 3D molecular geometries into ML features. The goal is to use this approach for high-throughput screening of molecular motifs that selectively bind CO₂ in presence of N₂. The idea is definitely interesting and new for quantum chemistry. My comments are given below.

1. The description of methodology is very unclear, especially that involving Figure 2 and other information in the SI. What the authors mean by 0D and 1D diagram etc in Fig. 2, when in text they talk about 1-D and 2-D images? How do they define dimensions? From SI, for a single atom they call it 0D, for two connected atoms they call it 1D although it is still 0th-order hole. In Fig. S3 they give a single value for each element, but in text they discuss 1D PIs formed by several elements (btw, is the number of involved atoms limited?). It is not clear either, why in Fig 2h the pixels are not symmetric, i.e. is it related to the orientation of a molecule H-Br?

2. Likely related to comment 1, the authors write in the SI that "... each xyz file that holds the Cartesian coordinates is first sorted by its elemental weight. This ensures that the persistence diagrams created for two different ordering of the same molecule are the same." How come that the order of elements affects the variance, if it is just some function of means of electronegativities of atoms involved in formation of 0th-order holes? And if the order is important, why is it not important to sort atoms with the same nuclear weight?

3. Probably the most critical comment. The authors did not convincingly demonstrate that their approach is size-consistent. Neither they performed numerical experiment that proves this claim, nor they provide clear mathematical proof. They did not show that training on smaller molecules will still provide reasonable ML predictions for larger molecules. They mention that the intensities on persistent surface (what is it anyway?) depend on number of specific connected components. It means that larger molecules will have higher intensities for the larger number of specific connected components. Then does the pixelation normalize these intensities? Then for example how ML model trained on a smaller oligomer will predict larger energy for a larger oligomer?

4. The results of their paper do not really support their claims that their method "generally outperforms other representations" as results obtained with BoB are often comparable or even better than results with persistent images.

5. The authors do not really describe how they chose the original 100 molecules. They refer to "the procedure described in Section 5.1." There are no numbered sections to start with (often, they send the reader to Sections with different numbers, which are non-existent).

6. They never explain in the manuscript what ML algorithm they used and why. Only in the SI the reader has a chance to learn that they used random forests, although it is not clear how it is more advantageous than other, more common in quantum chemical research ML algorithms. Generally, it is known that random forests are not as accurate for regression as other methods (Gaussian processes etc.). The poor choice of ML algorithm may explain rather high RMSE of 0.5 kcal/mol for noncovalent interactions.

7. Their final aim is to screen molecules with highest selectivity towards CO₂, however they never really discuss or use as their selection criterion the ratio between binding constants. It is not clear without such calculations how the three molecules they claim should be the best for separation, are really better than for example 4,5-diamino-1H-imidazol-2-ol, which is strongest binding to CO₂ and only has ca. -2.8 kcal/mol binding to N₂.

The manuscript has also numerous minor issues:

8. No energies and geometries of N₂ and CO₂ in SI, so no binding energy can be calculated.

9. a ML -> 'an ML'

10. greenhouses gases -> greenhouse gases

11. References for the first sentence and for "At an industrial level, liquid amine-based solvents are used for separation and capture of CO₂ via chemisorption, but the solvent regeneration step is an

energy intensive process."

12. "However, the number of potential CO₂-philic groups is intractably large, which leads to an excessive study of such systems with accurate ab initio methods" should be moved one sentence up.

13. "materialsørskov2013ørskov2013ørskov201332? -36" ???

14. "For chemical applications, this occurs through molecular representations, which are the featurization of molecular compounds from their three-dimensional structure into a vector of values."

There are other representations possible not based on 3D structure, e.g. SMILES.

15. 'intersections leads' -> 'intersections lead'

16. 'eg.' -> e.g.

17. 'eq.' -> e.g.

18. 'would be prohibit' -> 'would be prohibitive'

19. In each iteration of the active learning, do you choose next 40 molecules not present in the previous iteration?

20. 'this strategy allows ***the the*** model to ***encounter*** these rare instances within the training set' -> 'this strategy allows the model to account for these rare instances within the training set'

21. 'cross-fold validation errors' -> '10-fold cross-validation errors'

22. 'that that' -> 'that'

We would like to thank all reviewers for their feedback which substantially improved our manuscript. We have revised the manuscript based on *all* their comments. In this document we provide a point-by-point response to each comment raised (blue font), while excerpts from the text are shown when necessary with red font. Changes and/or additions in the actual manuscript are also shown with blue font.

Reviewer #2 (Remarks to the Author):

The authors present a molecular representation, which they consider to be easy to calculate and of the same size across all molecules (size-consistent). The authors apply it to train the CO₂/N₂ interaction energies of organic molecules within the GDB-9 set. The authors claim that this representation generally outperforms other representations.

The simplicity of the descriptor is very appealing and it appears to provide performance that is on par with the, by now outdated, coulomb matrix and BOB descriptors, but I do not see how the claim of 'generally outperforming other representations' is sufficiently substantiated in the manuscript.

Thank you very much for the support and interest in our work as well as the great points that you raised. Please find our response below and we hope that it alleviates any concerns.

-Within the error bars provided in Figure 3, BOB and PI show the same performance for a very small set of 100 interaction energies. I do not find this convincing. For such a small set of molecules, the authors could have systematically assessed a number of more state-of-the-art representations, such as FCHL, Behler-Parrinello, SOAP, or SchNet (not a descriptor per se, but a comparable approach). This would also have provided an opportunity to showcase the computational efficiency of PI, which is emphasized in the manuscript, but not quantified.

We have tested two additional state-of-the-art representations (SOAP, FCHL). The FCHL representation was investigated, and found not to increase the accuracy of the predictions (RMSE 0.50 ± 0.10 kcal/mol). We appreciate your recommendation for SOAP, as it provided good accuracy (0.41 ± 0.11 kcal/mol) comparable to our PI method (0.44 ± 0.09 kcal/mol, as explained later), with the caveat that its computational cost was approximately 40x greater than our method. In order to provide a fair comparison between BoBs, SOAP, FCHL and PI, we have performed a thorough study where we tested different learners, and optimized every representation/learner pair for minimizing their errors. Based on these recommendations, we have found that PI with kernel ridge regression and a Laplacian kernel provide an excellent tradeoff between computational cost and accuracy (0.44 ± 0.09 kcal/mol). A detailed analysis, which includes errors, timings, and optimized hyperparameters, has been added in the revised Supplementary Information (SI, Sections S7-S8), and the section "Performance of Persistence Images as Molecular Representations" of the manuscript has been rewritten (pages 10-12).

Here, we demonstrate the performance... yielding a higher confidence in the predictions.

The Behler-Parrinello and SchNet schemes are not suitable molecular representations for this application since they map atomic contributions, E^i to the total system energy E (or forces) via $E = \sum_i E^i$. While it would be possible to try to learn the total electronic energy, E , for the interacting systems, this training process would be tedious and require learning an entire potential energy surface for the interacting systems and isolated functional units. Herein, we avoid having to explicitly determine the CO₂ and N₂ functional unit-gas potential energy surface, as this is a time-demanding and computationally undesirable process. Because our training structures use only the bare functional unit, the total electronic energy expressed in atomic contributions is not a suitable representation for our application.

-The training protocol based on a series of active learning steps is well designed and seems appropriate for the envisioned application. As it only requires a small number of molecules, I wonder why the authors have not also trained a model for CM, BOB and PI to compare the methods within Figure 6. This would have provided for a more robust benchmark.

We agree with the reviewer that such a comparison will be extremely fruitful for the readers. We have screened the GDB-9 database with the four molecular representations (CM, BoB, SOAP and PI) and their corresponding optimal learners. We did not include the FCHL since it did not increase the accuracy. Figure 6 has now the plots from all schemes. The main conclusions are:

1. PI provided the most consistent results, without false negative cases which were present in CM, BoB (underestimated CO₂ interaction energies) or SOAP (incorrect stronger N₂ than CO₂ interaction energies).
2. All methods identified the strongest CO₂-philic group as the top hit, except SOAP.
3. The top-5 cases predicted by SOAP have more than 1 kcal/mol deviation from the “true” DFT energies (Table S13). On the contrary, PI predictions from the same five cases were significantly closer to the DFT values (~0.30 kcal/mol, Table S13).

This analysis has been added in the revised manuscript (Pages 16-18).

For comparison, the database was screened with... on an Intel®i5-4278U processor.

- I worry about the significance of the findings in Figure 6 and the CO₂/N₂ interaction energy ratios that are discussed. All N₂ interaction energies fall within a range of 1 kcal/mol, which is somewhat below the trust radius of PBE0-D3. The authors select PBE0-D3 based on a benchmark of CO₂ interaction energies against coupled-cluster results, but it is not clear from the manuscript or SI if this level of confidence also applies to N₂.

We certainly agree with your conclusions. As verification, we ran CCSD(F12)(T)/cc-pVTZ calculations on several of the systems (3,12,18,29) and found RMSE of 0.27 and 0.29 kcal/mol for CO₂ and N₂, respectively. This specific level of theory provides deviations for CO₂ and N₂ intrafraction energies are of the same magnitude. Since interaction energies of N₂ are within a range of ~1 kcal/mol, values for CO₂ hold more significance and thus, N₂ prediction were not used as a criterion for the expansion of the dataset. However, since the error for DFT is approximately within the error of the ML method we believe these findings are still of significant value.

- It is not clear what the benefit of a 'size-consistent' descriptor is. As the authors state, all descriptors can be padded to be of consistent size for molecules of different composition. The authors do not provide evidence for cases in which there would be a material benefit to size-consistency. How is this property particularly beneficial in the proof-of-principle application that they discuss? Figure 2 also shows that the range that is covered in the pixelated diagrams that correspond to the final descriptor depends on the size of the molecule (no. of births and level of persistence can vary). This means that the size of the Birth/Persistence axis is controlled by the largest and most complex molecule in the dataset. One could consider this to be a form of zero-padding when the same range is applied to smaller/less complex molecules.

Collins *et al.* (<https://doi.org/10.1063/1.5020441>) have argued that a constant-size representation is beneficial because it can be easier generalized from smaller to larger systems. Adopting this point of view, we changed the “size-consistent” confusing argument to “constant-size representation”. We have also added a new section in the SI document (Section S5) where we present a numerical example between a small molecule (anisole) and a large organic oligocycle (4-tert-butylcalixarene). We believe it is now evident that the persistence images of a small and a large molecule have the same size, while extensive padding of a non-constant-size representation like CM/BoB is unavoidable. The benefits of such a fixed-vector representation will be the topic of a follow-up study since the aim of this manuscript was to provide a proof-of-concept of the applicability of chemically-driven persistent homology.

We have rewritten the corresponding section of the manuscript (pages 10):

Another important feature is related to the dimensionality of the PI molecular representation which remains almost unaffected by the molecular size. For example, for a molecule like anisole that is composed by 15 atoms, a PD that has $3\text{\AA} \times 3\text{\AA}$ was generated. The equivalent PD for a significantly larger molecule such as the tert-butylcalixarene (105 atoms) is of comparable size ($4\text{\AA} \times 4\text{\AA}$). A detailed analysis is given in Section S5 of the Supplementary Information. As it was mentioned in the introduction, such a constant-size representation is desirable for many chemical applications.

Reviewer #3 (Remarks to the Author):

The manuscript introduces persistent homology-based method for transforming 3D molecular geometries into ML features. The goal is to use this approach for high-throughput screening of molecular motifs that selectively bind CO₂ in presence of N₂. The idea is definitely interesting and new for quantum chemistry.

Thank you very much for the support and interest in our work as well as the great points that you raised. Please find our response below and we hope that it alleviates any concerns.

My comments are given below.

1. The description of methodology is very unclear, especially that involving Figure 2 and other

information in the SI. What the authors mean by 0D and 1D diagram etc in Fig. 2, when in text they talk about 1-D and 2-D images? How do they define dimensions? From SI, for a single atom they call it 0D, for two connected atoms they call it 1D although it is still 0th-order hole.

We apologize for the confusing arguments on the dimensions and the corresponding unclear definitions.

In order to generate a persistence diagram, we basically transform the molecular compound to a simplicial complex (by increasing the radii of spheres around each atom), and we track the birth and the persistence of homological features such as connected components (0-dimensional homological features) and holes (1-dimensional homological features). This is what we meant by 0D vs 1D diagrams. This process is described in Figure 1.

Now, naively speaking, a simplicial complex is a union (with certain rules) of simple pieces. An example of a simple piece is a vertex and an edge, which are considered a 0D and 1D simple pieces respectively. This is why we called the atoms 0D, and when we connected them 1D (we actually referred to the edge between them). By the same token, when two atoms are connected then they form a connected component, which is a 0-dimensional homological feature. Essentially think of a homological feature as a cycle, and thus a connected component is a 0th order cycle.

Moreover, when one focuses on persistence diagrams that describe connected components, their birth is at 0 since the algorithm starts at radius of spheres 0. To that end they form a column vector and this is why we called the associated persistence image (PI) a 1-D image. On other hand, when one focuses on persistence diagrams that describe holes, their birth could be anywhere on the plane, and thus, the associated PIs form a 2-column matrix. This is why we called the associated PI a 2-D image.

All in all, we recognize the fact that all of the aforementioned could cause a severe confusion to an uninitiated reader. We have simplified the description and we chose a universal language across the manuscript. Indeed, we avoid any mention of confusing dimensions, but we rather clearly lay out when we mean connected components and/or holes as well as their associated generated persistence diagrams and the PI of molecules. We hope that these edits alleviate any further confusion.

In Fig. S3 they give a single value for each element, but in text they discuss 1D PIs formed by several elements (btw, is the number of involved atoms limited?).

Figure S3 had the individual values based on the electronegativity of each atom. We have updated the SI and in Section S4 we explain in detail how the PIs are generated. We believe that the revised SI provides a clear description of our computational process.

It is not clear either, why in Fig 2h the pixels are not symmetric, i.e. is it related to the orientation of a molecule H-Br?

About Figure 2(h), the pixels are not symmetric because of the location of the mean of the Gaussian with respect to the grid used.

2. Likely related to comment 1, the authors write in the SI that "... each xyz file that holds the Cartesian coordinates is first sorted by its elemental weight. This ensures that the persistence diagrams created for two different ordering of the same molecule are the same." How come that the order of elements affects the variance, if it is just some function of means of electronegativities of atoms involved in formation of 0th-order holes? And if the order is important, why is it not important to sort atoms with the same nuclear weight?

Thank you for bringing this up. We have updated the algorithm by not considering ordering by the Elder Rule. However, in our new implementation, when two spheres intersect, we take the electronegativity difference of the two corresponding atoms (centers of the spheres) as pointed out in Equation (S1) of Section S4.

3. Probably the most critical comment. The authors did not convincingly demonstrated that their approach is size-consistent. Neither they performed numerical experiment that proves this claim, nor they provide clear mathematical proof.

Collins *et al.* (<https://doi.org/10.1063/1.5020441>) have argued that a constant-size representation is beneficial because it can be easier generalized from smaller to larger systems. Adopting this point of view, we changed the "size-consistent" confusing argument to "constant-size representation". We have also added a new section in the SI document (Section S5) where we present a numerical example between a small molecule (anisole) and a large organic oligocycle (4-tert-butylcalixarene). We believe it is now evident that the persistence images of a small and a large molecule have the same size, while extensive padding of a non-constant-size representation like CM/BoB is unavoidable. The benefits of such a fixed-vector representation will be the topic of a follow-up study since the aim of this manuscript was to provide a proof-of-concept of the applicability of chemically-driven persistent homology.

We have rewritten the corresponding section of the manuscript (pages 10):

Another important feature is related to the dimensionality of the PI molecular representation which remains almost unaffected by the molecular size. For example, for a molecule like anisole that is composed by 15 atoms, a PD that has $3\text{\AA} \times 3\text{\AA}$ was generated. The equivalent PD for a significantly larger molecule such as the tert-butylcalixarene (105 atoms) is of comparable size ($4\text{\AA} \times 4\text{\AA}$). A detailed analysis is given in Section S5 of the Supplementary Information. As it was mentioned in the introduction, such a constant-size representation is desirable for many chemical applications.

They did not show that training on smaller molecules will still provide reasonable ML predictions for larger molecules. They mention that the intensities on persistent surface (what is it anyway?) depend on number of specific connected components. It means that larger molecules will have higher intensities for the larger number of specific connected components. Then does the

pixelation normalizes these intensities? Then for example how ML model trained on a smaller oligomer will predict larger energy for a larger oligomer?

The persistence surface consists of a sum of Gaussians that are each centered at points of the persistence diagram. We have removed the word “surface” in order to avoid further confusion. Indeed, larger molecules will produce higher intensity pixels which is related to the distance from the pixel of the PI to the points of the PD. The closer the pixel is to point(s), the larger the intensity value. In our implementation, pixelation does not normalize the intensities, but it is an interesting idea that we will consider in the future, together with the prediction of properties of larger molecules from data obtained from smaller molecules. We have communicated this in the Discussion section (page 19):

For this type of chemical applications, additional features are taken into consideration, such as intensity normalization when a PI is generated from the corresponding PD and predictability of properties of larger molecules from data generated from smaller ones.

4. The results of their paper do not really support their claims that their method "generally outperforms other representations" as results obtained with BoB are often comparable or even better than results with persistent images.

In the revised manuscript, we have expanded our discussion to other molecular representations (e.g. FCHL, SOAP). We have optimized the choice of the learner for each method and we have screened the GDB-9 database for each method (see Figure 6). A detailed analysis has been added in the SI (Section S7), and the section “Performance of Persistence Images as Molecular Representations” of the manuscript has been rewritten (pages 10-12).

Here, we demonstrate the performance... yielding a higher confidence in the predictions.

5. The authors do not really describe how they chose the original 100 molecules. They refer to "the procedure described in Section 5.1." There are no numbered sections to start with (often, they send the reader to Sections with different numbers, which are non-existent).

We apologize for the erroneous reference to *Section 5.1*. The procedure is described in the SI, Section S1.

6. They never explain in the manuscript what ML algorithm they used and why. Only in the SI the reader has a chance to learn that they used random forests, although it is not clear how is it more advantageous than other, more common in quantum chemical research ML algorithms. Generally, it is known that random forests are not as accurate for regression as other methods (Gaussian processes etc.). The poor choice of ML algorithm may explain rather high RMSE of 0.5 kcal/mol for noncovalent interactions.

We would like to thank the reviewer for this comment, since it motivated us to optimize the ML part of all molecular representations considered in this work (CM, BoB, FCHL, SOAP, and PI). Indeed, the RMSE dropped to ~0.4 kcal/mol for SOAP and PI. A detailed analysis has been added in the SI (Section S7) and parts of the last two sections of the manuscript has been rewritten (pages 10-18).

7. Their final aim is to screen molecules with highest selectivity towards CO₂, however they never really discuss or use as their selection criterion the ratio between binding constants. It is not clear without such calculations how the three molecules they claim should be the best for separation, are really better than for example 4,5-diamino-1H-imidazol-2-ol, which is strongest binding to CO₂ and only has ca. -2.8 kcal/mol binding to N₂.

Thanks for this comment. We did not intend to suggest the other three molecules were superior to 4,5-diamino-1H-imidazol-2-ol, which was also found during the active learning process. That discussion has been removed in the revised manuscript and that section is summarized as:

Ideally, increasing CO₂ selectivity entails maximizing CO₂ interaction while attempting to reduce N₂ interaction.

The manuscript has also numerous minor issues:

8. No energies and geometries of N₂ and CO₂ in SI, so no binding energy can be calculated.
Corrected, thank you.

9. a ML -> 'an ML'
Corrected.

10. greenhouses gases -> greenhouse gases
Corrected.

11. References for the first sentence and for "At an industrial level, liquid amine-based solvents are used for separation and capture of CO₂ via chemisorption, but the solvent regeneration step is an energy intensive process."
Thank you, this has been addressed.

12. "However, the number of potential CO₂-philic groups is intractably large, which leads to an excessive study of such systems with accurate ab initio methods" should be moved one sentence up.
Completed

13. "materialsørskov2013ørskov2013ørskov201332? -36" ???
Fixed

14. "For chemical applications, this occurs through molecular representations, which are the featurization of molecular compounds from their three-dimensional structure into a vector of values." There are other representations possible not based on 3D structure, e.g. SMILES.

Thank you for this clarification. We have changed the text from "three-dimensional structure" to "molecular structure"

15. 'intersections leads' -> 'intersections lead'

Corrected.

16. 'eg.' -> e.g.

Corrected.

17. 'eq.' -> e.g.

Corrected.

18. 'would be prohibit' -> 'would be prohibitive'

Corrected.

19. In each iteration of the active learning, do you choose next 40 molecules not present in the previous iteration?

Correct, we select the next 40 molecules that do not present duplication of structures previously added.

20. 'this strategy allows ***the the*** model to ***encounter*** these rare instances within the training set' -> 'this strategy allows the model to account for these rare instances within the training set'

Thank you, this change has been made.

21. 'cross-fold validation errors' -> '10-fold cross-validation errors'

Added.

22. 'that that' -> 'that'

Corrected.

Reviewers' comments:

Reviewer #2 (Remarks to the Author):

The authors have provided substantive changes to the manuscript and addressed most of the raised points in detail, which I appreciate. This certainly helps to alleviate any concerns on the validity and significance of the performance of the method and the presented application. The method is interesting, the work is thorough, so it should be published.

Regarding the question if the paper will influence thinking in the field, I am not sure.

The presented homology feature is clever and the concept underutilized in chemistry, but it does not provide for dramatic prediction improvement compared to existing features. I am still not able to see what the killer application is for which this feature will outshine everything else. Catalytic high-throughput screening is probably not that application as all other (efficient and computationally costly) features seem to do reasonably well (at least within the accuracy of the underlying electronic structure methods). Maybe the authors could provide more outlook on that.

One more very minor point:

Rewording size-consistent as constant-size representation will surely help to avoid confusion, but is PD really of constant size? The example that is provided for small and large molecules deliver representations that are very similar in size (3x3A) vs. (4x4A), but not identical if the pixel spacing is the same or there is no padding (which would be absolutely fine if used). What is the actual length of the contracted feature vector? For very large molecules, there are many more features, so the pixel density needs to be increased as well or the image needs to be sampled and processed somehow differently. This should be explained better, if it is to serve as a key advantage of the method.

Reviewer #3 (Remarks to the Author):

Revised manuscript is an improvement over the previous version as they improved their ML algorithm and compared to additional existing approaches. They considered most of criticism, but not all or not appropriately.

The authors highlight smaller error bars (not energies) for PI than for SOAP for CO₂-binding energy, but if you look at N₂-binding energies then all models are practically the same. Is there any reason for this?

The modified Figure 6 is not appropriate because it is based on a biased approach. The training set was selected via active learning that was run only with PI, not with other ML approaches. Their results for the initial 100 training points do not show so big difference between methods. More appropriate would be to generate plots like Figure 6 after each iteration (at least for the initial 100 points) and run each iteration with the corresponding approach (CM, BoB, SOAP).

It would be good if the authors could provide a concise figure with molecular groups and discussion addressing their declared aim "to screen a large molecular database in order to discover molecular groups that show a stronger affinity for CO₂ interaction over N₂". Making unbiased comparison (see previous comment) with different methods in this regard would be also helpful

It looks strange that after the authors made changes to the ML model, it did not change anything in

the discussion of the active learning procedure, i.e. in both revisions they write e.g. "two anionic structures were found during the third step", "mean and median both at -5.38 kcal/mol.", "the second iterations, at -5.43 and -5.26".

The authors changed from the claim that their method is size-consistent to that their method is constant-size, but then for anisole the size of PD is 3x3, while for tert-butylcalixarene it is 4x4. Yes, they similar, but apparently not constant. So their model is neither size-consistent nor constant-size.

In the rebuttal letter they make claim that SchNet and Behler-Parrinello are not suitable, because they represent energy as atomic contributions, but ML models with SOAP are typically considered to do just that (see Ref. 60 for example). If the authors did not do this in this manuscript, then there is a danger that SOAP was not really used properly. It still gives rather good results.

"In generating molecular representations from a molecular structure, the degrees of freedom of the structure are inherently reduced, where many geometric properties are truncated and passed to the ML model." This claim is dubious. For examples the authors given (Coulomb matrix etc.) the degrees of freedom measured by number of features for a molecule actually increases.

In rebuttal letter, the authors write that "the pixels are not symmetric because of the location of the mean of the Gaussian with respect to the grid used.", this point is still not clear, in the SI one can find "Gaussian (birth, persistence)+N(0, $\sigma^2 \cdot SF_{ij}$)", i.e. they use normal distribution with zero mean...

The authors still do not describe how they chose the original 100 molecules, Section S1 they refer too does not say anything. Were they chosen randomly from QM9?

Authors again refer to some non-existing sections, e.g. 'Section 3.2', even after pointing out this type of problems in the comments to the original submission. I do not list here all the sections (important note, because authors only fixed the Section I mentioned).

I still could not find energies and geometries of N2 and CO2 in SI or on their gitlab, although authors claimed that they added them...

Fix:

- * The authors still did not correct one minor issue, although in rebuttal letter they claimed they did: "However, the number of potential CO2-philic groups is intractably large, which leads to an excessive study of such systems with accurate ab initio methods" should be moved one sentence up. Otherwise it does not make any sense.
- * 'depend on the to distances'
- * Please do not invent terms for 'Laplacian kernel ridge regression' and 'linear ridge regression'. They are kernel ridge regression with the Laplacian or linear kernel.
- * Figure S12 -> Table S12.

We would like to thank all reviewers for their positive comments and feedback. We provide a second revision on our manuscript based on *all* their comments. In this document we provide a point-by-point response to each comment raised (blue font), while excerpts from the text are shown when necessary with red font. Changes and/or additions in the actual manuscript are also shown with blue font.

Reviewer #2 (Remarks to the Author):

The authors have provided substantive changes to the manuscript and addressed most of the raised points in detail, which I appreciate. This certainly helps to alleviate any concerns on the validity and significance of the performance of the method and the presented application. The method is interesting, the work is thorough, so it should be published.

Thank you for your constructive comments and your support.

Regarding the question if the paper will influence thinking in the field, I am not sure. The presented homology feature is clever and the concept underutilized in chemistry, but it does not provide for dramatic prediction improvement compared to existing features. I am still not able to see what the killer application is for which this feature will outshine everything else. Catalytic high-throughput screening is probably not that application as all other (efficient and computationally costly) features seem to do reasonably well (at least within the accuracy of the underlying electronic structure methods). Maybe the authors could provide more outlook on that. This is a great and inspiring comment. We have computed the persistence diagram of the main protease of COVID-19 in complex with an inhibitor N3, which is composed by 2500 non-hydrogen atoms (crystal structure published online on March 29, <https://www.biorxiv.org/content/10.1101/2020.02.26.964882v3>). The PI has dimensions of $6\text{\AA}\times 6\text{\AA}$, and we needed only 131 seconds to generate it. For SOAP, 972 features are needed per atom (2500×972), which means that the time spent for the REMake kernel step would be prohibitively large. Thus, we believe that the PI representation can be used in a larger number of chemical and biological applications.

One more very minor point:

Rewording size-consistent as constant-size representation will surely help to avoid confusion, but is PD really of constant size? The example that is provided for small and large molecules deliver representations that are very similar in size ($3\times 3\text{\AA}$) vs. ($4\times 4\text{\AA}$), but not identical if the pixel spacing is the same or there is no padding (which would be absolutely fine if used).

We realize that both terms that we have been using (size-consistent, constant-size) are confusing, even if the representation has a constant size for a given dataset, and not for every possible molecule. For that reason, we have removed the “constant-size” term from both manuscript and SI and we have introduced a short discussion on the similar size representation, i.e. the same order with respect to the system size, when we compare PI with BoB and CM (see last paragraph in page 10 “Another important feature...”).

What is the actual length of the contracted feature vector? For very large molecules, there are many more features, so the pixel density needs to be increased as well or the image needs to be

sampled and processed somehow differently. This should be explained better, if it is to serve as a key advantage of the method.

In this work, a fine grid was used. Precisely, the size of the medium-size molecule (4Åx4Å PD) has a 200x200 grid size, while the one for the large enzyme (6Åx6Å PD) has a 300x300 grid size. We have not performed an analysis how the grid depends as a function of the size of the structure when we move from small molecules to large enzymes, since it is beyond the scope of the current manuscript. However, we believe that this should not be an issue given that points on a PD are extremely close to each other provide similar topological information. Again, this is currently a topic of active research in our groups.

Reviewer #3 (Remarks to the Author):

Revised manuscript is an improvement over the previous version as they improved their ML algorithm and compared to additional existing approaches. They considered most of criticism, but not all or not appropriately.

Thank you for your constructive comments and your support.

The authors highlight smaller error bars (not energies) for PI than for SOAP for CO₂-binding energy, but if you look at N₂-binding energies then all models are practically the same. Is there any reason for this?

The reason in the small distribution of N₂ interaction energies, which range between -1.6 and -2.6 kcal/mol, while for CO₂, the range is significantly larger (-2.0 to -7.0 kcal/mol). This is mentioned in the caption of Figure S4.

The modified Figure 6 is not appropriate because it is based on a biased approach. The training set was selected via active learning that was run only with PI, not with other ML approaches. Their results for the initial 100 training points do not show so big difference between methods. More appropriate would be to generate plots like Figure 6 after each iteration (at least for the initial 100 points) and run each iteration with the corresponding approach (CM, BoB, SOAP).

We agree that for an extensive comparison between the models, active learning should be performed for each case. We have performed the same active learning steps for all representations (CM, BoB, SOAP and PI) and we used optimized ML model that yielded the lowest errors for each method, as it was found in the extensive testing with the 100-molecule dataset (Section S6). Thus, the KRR (Laplacian kernel) was used for CM and PI, GRP for BoB, and KRR (linear kernel) for SOAP. For that reason, we had to perform more than 28,000 DFT calculations based on the computational protocol discussed in Sections S4 and S5. The reason that we had to perform this large number of DFT calculations is that each molecular representation identifies different top-40 hits per active learning step. We found some overlap between the third steps of PI and SOAP.

We have updated the manuscript and the discussion between the four molecular representations is now based on the optimized models. Here, we summarize our main conclusions extracted from this extended:

1. CO₂ and N₂ interaction energies predicted by PI with the KRR (Laplacian) learner are similar to those reported in the previous revised version.

2. Revised SOAP showed similar accuracy with the previous version but a) it identified many molecules with low CO₂ binding (in a few instances it predicted erroneously positive, repulsive interaction energies), b) it needed more than 30 hours to screen GDB-9 on a 20-core system (PI needed 637 seconds on an 8-core processor), and c) it provided low mean and median for each step of the process (Table S12).
3. GPR with BoB yielded unphysical results. On the contrary, BoB with the KRR (linear learner gave reliable CO₂/N₂ distributions similar to the other methods (Section S8).
4. The active learning steps performed with the CM and BoB provided poorer top hits, as it is shown in Figures 5 and 6. Ironically, original data from PI (previous revised version) made these two representations to provide more reliable CO₂/N₂ predictions. We have included a short discussion about this in Section S8.
5. Violin plots nicely explain the progress of the four molecular representations during the active learning process, where PI (and partially SOAP) were able to identify molecules with higher CO₂-philicity (i.e. rare events within the GDB-9 database) in only three active learning steps. For that reason, we have included in the revised manuscript a plot per method (Figure 5).

It would be good if the authors could provide a concise figure with molecular groups and discussion addressing their declared aim "to screen a large molecular database in order to discover molecular groups that show a stronger affinity for CO₂ interaction over N₂". Making unbiased comparison (see previous comment) with different methods in this regard would be also helpful

Thank you for your suggestion. We did compare the top hits from the four new, optimized models (see previous comment). We believe that an extensive table with all molecular groups (eg. hydroxy, methoxy, methyl, amino groups etc.) that appear in GDB-9 database is beyond the scope of this manuscript. Previous studies (eg. ChemPhysChem 2, 374, Phys. Chem. Chem. Phys. 17, 10925, Int. J. Quantum Chem. 113, 2261, J. Chem. Phys. 134, 034301, Environ. Sci. Technol. 45, 8582) have examined the interaction of CO₂ with a small number of molecules (10-50). Here, we extend these individual studies with a systematic screening of 133,000 organic molecules and we were able to identify new molecular species that interact stronger with CO₂.

It looks strange that after the authors made changes to the ML model, it did not change anything in the discussion of the active learning procedure, i.e. in both revisions they write e.g. "two anionic structures were found during the third step", "mean and median both at -5.38 kcal/mol.", "the second iterations, at -5.43 and -5.26".

The screening process has now been repeated (see previous comment) and the active learning procedure was updated.

The authors changed from the claim that their method is size-consistent to that their method is constant-size, but then for anisole the size of PD is 3x3, while for tert-butylcalixarene it is 4x4. Yes, they similar, but apparently not constant. So their model is neither size-consistent nor constant-size.

We realize that both terms that we have been using (size-consistent, constant-size) are confusing, even if the representation has a constant size for a given dataset, and not for every possible molecule. For that reason, we have removed the "constant-size" term from both manuscript and SI and we have introduced a short discussion on the similar size representation, i.e. the same

order with respect to the system size, when we compare PI with BoB and CM (see last paragraph in page 10 “Another important feature...”).

In the rebuttal letter they make claim that SchNet and Behler-Parrinello are not suitable, because they represent energy as atomic contributions, but ML models with SOAP are typically considered to do just that (see Ref. 60 for example). If the authors did not do this in this manuscript, then there is a danger that SOAP was not really used properly. It still gives rather good results.

We believe we have used SOAP properly together with the REMatch kernel, as it is described in Section S7, in line with the tutorial (<https://singroup.github.io/dscribe/tutorials/kernels.html>).

"In generating molecular representations from a molecular structure, the degrees of freedom of the structure are inherently reduced, where many geometric properties are truncated and passed to the ML model." This claim is dubious. For examples the authors given (Coulomb matrix etc.) the degrees of freedom measured by number of features for a molecule actually increases. We agree with the reviewer. We have deleted this sentence since it brings confusion and since it does not affect the analysis.

In rebuttal letter, the authors write that "the pixels are not symmetric because of the location of the mean of the Gaussian with respect to the grid used.", this point is still not clear, in the SI one can find "Gaussian (birth, persistence)+N(0, $\sigma^2 \cdot SF_{ij}$)", i.e. they use normal distribution with zero mean...

The mean of the Gaussian changes according to (birth, persistence) given that it is an additive Gaussian model. We have added one extra sentence on the SI to clarify that.

The authors still do not describe how they chose the original 100 molecules, Section S1 they refer too does not say anything. Were they chosen randomly from QM9?

The initial structures were not chosen from QM9, but their choice was based on the observation that aromatic heterocycles have strong CO₂ affinity. The word “aromatic” has been added on Section S1.

Authors again refer to some non-existing sections, e.g. 'Section 3.2', even after pointing out this type of problems in the comments to the original submission. I do not list here all the sections (important note, because authors only fixed the Section I mentioned).

Thank you for highlighting this mistake. We have removed the non-existing sections in the revised manuscript.

I still could not find energies and geometries of N₂ and CO₂ in SI or on their gitlab, although authors claimed that they added them...

We would like to apologize for omitting the geometries of CO₂ and N₂. We have now added them both in the SI (page S-71) and in the online repository.

Fix:

* The authors still did not correct one minor issue, although in rebuttal letter they claimed they did: "However, the number of potential CO₂-philic groups is intractably large, which leads to an

excessive study of such systems with accurate ab initio methods" should be moved one sentence up. Otherwise it does not make any sense.

We believe that the current ordering makes more sense. No changes were made.

* 'depend on the to distances'

Fixed

* Please do not invent terms for 'Laplacian kernel ridge regression' and 'linear ridge regression'. They are kernel ridge regression with the Laplacian or linear kernel.

We have rewritten this sentence as:

The most accurate models were with kernel ridge regression (Laplacian kernel, 0.44 kcal/mol) and SOAP with kernel ridge regression (linear kernel, 0.41 kcal/mol),

* Figure S12 -> Table S12.

Thanks.

REVIEWERS' COMMENTS:

Reviewer #3 (Remarks to the Author):

In the revised manuscript the authors provided a nice proof of the advantage of their unique method over other existing machine learning approaches for the high-throughput screening of molecules binding stronger to CO₂ than to N₂.